# Inhibition of Succinate Dehydrogenase by Pesticides (SDHIs) and Energy Metabolism

**DOI:** 10.3390/ijms24044045

**Published:** 2023-02-17

**Authors:** Frederic Bouillaud

**Affiliations:** Institut Cochin, INSERM, CNRS, Université Paris Cité, 75014 Paris, France; frederic.bouillaud@inserm.fr

**Keywords:** mitochondria, tricarboxylic acid cycle, metabolic syndrome, liver, redox state

## Abstract

Succinate dehydrogenase (SDH) is one of the enzymes of the tricarboxylic acid cycle (Krebs cycle) and complex II of the mitochondrial respiratory chain. A class of fungicides (SDHIs) targets the complex II reaction in the SDH. A large number of those in use have been shown to inhibit SDH in other phyla, including humans. This raises questions about possible effects on human health and non-target organisms in the environment. The present document will address metabolic consequences in mammals; it is neither a review on SDH nor is it about the toxicology of SDHIs. Most clinically relevant observations are linked to a severe decrease in SDH activity. Here we shall examine the mechanisms for compensating a loss of SDH activity and their possible weaknesses or adverse consequences. It can be expected that a mild inhibition of SDH will be compensated by the kinetic properties of this enzyme, but this implies a proportionate increase in succinate concentration. This would be relevant for succinate signaling and epigenetics (not reviewed here). With regard to metabolism, exposure of the liver to SDHIs would increase the risk for non-alcoholic fatty liver disease (NAFLD). Higher levels of inhibition may be compensated by modification of metabolic fluxes with net production of succinate. SDHIs are much more soluble in lipids than in water; consequently, a different diet composition between laboratory animals and humans is expected to influence their absorption.

## 1. Introduction

SDHIs constitute a class of pesticides to fight against fungi. This represents roughly a dozen different molecules (Figure 1) sharing the property to inhibit the succinate dehydrogenase (SDH), an enzyme implicated in carbon metabolism and cellular respiration. SDH is also known as complex II of the mitochondrial respiratory chain (MRC). Actually, SDH oxidizes the succinate into fumarate to feed mitochondrial respiration. Consequently, the inhibition of SDH is expected to lower cellular consumption of succinate (that may accumulate) and to impair cellular respiration. Inhibition of SDH is most likely to ground the fungicide effect of SDHIs, and acquired resistance to SDHI in pests correlates with mutations in SDH genes [1]. Experimental evidences in vitro indicate, from the earliest reports with carboxin [2], that SDHI inhibited the SDH of mammals. This has been reexamined recently with eight SDHIs actually in use [3] (Figure 1). This recent report boosted the controversy about the use of these pesticides because it could be considered that extended use of a toxicant to human mitochondrial oxidative metabolism would inevitably result in deleterious consequences in the long term, and/or in more sensitive individuals. These alterations could be anticipated from existing knowledge about pathologies resulting from genetic inactivation of SDH, and from the scientific literature discussing other inhibitors of SDH. At this point it should be highlighted that few mechanistic studies on the impact of different SDHI pesticides on SDH and mitochondrial respiration are presently available. With regard to pesticides acting through inhibition of mitochondrial respiration and the apparent resistance of human beings in contrast with target species, a past example is provided by the prolonged use and eventually banishment of the complex I inhibitor rotenone [4,5]. The question is, therefore, the risk of inhibition of human SDH following accidental or chronic (contamination of food) exposure to these SDHIs and possible consequences. Toxicological studies in rodent or other mammals were required for authorization of use of SDHIs. The blockade of cellular respiration in mammals, as with cyanide, results in death within a few minutes, if not seconds, and SDHIs would have been classified as highly toxic if that were the case. While these toxicological studies are not made public, they allowed the definition of tolerable exposure levels (Figure 1). The existing literature shows that other inhibitors of SDH result in the poisoning of mammalian SDH, with neurological consequences that become severe (irreversible lesions with neuronal death) when inhibition of SDH reaches 30–50% [6]. Genetic data obtained in SDHD-KO mice [7], and with alleles causing a loss of function of SDH in humans that are autosomal and recessive [8,9], indicate that SDH activity is required. Transient inhibition of cellular respiration is not inevitably deleterious in the mid-term, and transient ischemia, or inhibition of cellular respiration (preconditioning), including with an SDH inhibitor [10,11], triggers adaptive mechanisms that protect against a subsequent higher intensity event of the same nature. This opens the possibility that a given inhibitor of mitochondrial activity would show a biphasic effect with a rather positive side for low levels of exposure and deleterious consequences for the higher levels. Low and high may mean, here, dosage and/or duration. Finally, too much activity of the SDH enzyme might have deleterious consequences, and partial inhibition of SDH might be considered a therapeutic approach [12].

Unless the precaution principle outlined before is adopted, the situation appears intricate. A part of this complexity might be understood after a detailed examination of the cellular biochemistry of SDH, and this may help to interpret existing approaches or to envisage new experimental approaches to define the potential risks of exposure to SDHIs. The present document aims to do so about the implications of SDH inhibition for energy metabolism in mammals. The theoretical grounds used here relate to basic biochemistry and (over)simplified models. This document is not a comprehensive review about SDH or succinate. This is also not a comprehensive review on the toxicology of SDHIs, because the latter should examine all possible adverse effects of each SDHI and not only those anticipated from the common property of SDH inhibitors. An opposite path is presented here: from the properties of the target enzyme (SDH), a wide range of possible consequences is examined to propose those that appear most relevant.

## 2. SDH and Cellular Energy Metabolism

Because SDH participates in the cellular energy metabolism, the main characteristics of the latter will be briefly summarized before more detailed examination of SDH itself. Cellular energy expenditure is mainly supported by the hydrolysis of the last phosphate bond of adenosine triphosphate (ATP > ADP + Pi). ATP is regenerated continuously by the reverse reaction (phosphorylation of ADP). In mammals, this phosphorylation is the result of two processes: the mitochondrial respiration, or the lactic fermentation. The latter is actually efficient locally/transiently, but lactate is not significantly released as waste in the environment, and eventually mitochondrial respiration is the ATP provider in mammals. The respiration of an animal uses oxygen (O_2_) and releases carbon dioxide (CO_2_). It is pertinent to examine separately these two processes. CO_2_ comes from substrate oxidation (metabolism), which provides electrons to the oxidative phosphorylation (Oxphos) that reduces O_2_ into water (Figure 2). Metabolism (as defined above) includes a large number of enzymes, and few of them release ATP during carbon oxidation. However, Oxphos is quantitatively the main contributor to ATP regeneration. Oxphos is ensured by five enzymatic complexes and two electron shuttles (quinone and cytochrome-c), all located in the inner mitochondrial membrane. Complexes I-IV constitutes the mitochondrial respiratory chain (MRC) that catalyze successive steps of electron transfer from a donor to oxygen. Complexes I, III, and IV convert the energy of oxidation into a proton electrochemical gradient used by complex V; that is the ATP producing machine. Complex II is the SDH.

### Comparison between SDH/Complex II and the Other Mitochondrial Respiratory Complexes

The similarities, relations, and differences between SDH/complex II and other respiratory complexes are summarized below:Respiratory complexes are very large heteromeric assemblies (more than one million dalton) inserted in the mitochondrial inner membrane. This means a complex/long assembly process. A consequence is a long lifetime that allows integration and binding of inhibitors with time. The large majority of the subunits constituting these respiratory complexes are encoded by the nuclear genome, but in addition complexes I, III, IV and V contain a minority of subunits coded by the mitochondrial genome. In contrast, all subunits of complex II are coded by the nuclear genome.The oxidative phosphorylation mechanism follows Mitchell’s chemiosmotic theory, according to which proton gradient and movement explain the bioenergetic coupling (Figure 2). Accordingly, complexes I, III, IV, and V associate proton movement to another reaction: redox reactions to proton pumping in complexes I, III, and IV, or phosphorylation of ADP into ATP to proton reentry in complex V. In contrast, complex II does not catalyze proton movement and is uniquely a redox enzyme.Complexes I, II, III, and V could operate in the opposite direction depending on forces, which for complexes I, III, and V include the proton gradient (membrane potential). The proton gradient has no direct effect on complex II (SDH) that (in absence of inhibitors) is uniquely sensitive to the ratio of succinate/fumarate and to the reduction/oxidation state of quinone.Complexes I, III, and IV of the MRC accept electrons from electron shuttles (NAD, quinone, cytochrome c) and, therefore, the link with metabolic redox reactions is indirect. In contrast, the link between carbon metabolism and electron supply to the respiratory chain is direct and relies on electron transfer within the SDH itself.Oxidation of glucose and of fatty acids loads the largest part of the electrons on the NAD/NADH shuttle and complex I is, therefore, the main entry for electrons in the respiratory chain. However, complex II and the electron-transferring flavoprotein complex reoxidizing FADH_2_ from beta oxidation of fatty acids and a few others yield electrons directly to quinone with two consequences: (i) the yield in ATP of the electron transfer to oxygen is lower (1.6 ATP) per two electrons (oxygen atom) instead of 2.7 with complex I, and, (ii) quinone appears to be the point of convergence for the entry of electrons, with only one exit (complex III) and, consequently, competition between electron donors could take place, notably between complex I and II.

## 3. The Succinate Dehydrogenase or Mitochondrial Complex II

SDH catalyzes the oxidation of succinate into fumarate according to the following equation:HOOC-CH_2_-CH_2_-COOH ➔ HOOC-CH = CH-COOH + 2H^+^ + 2*e*^−^

The two protons and electrons correspond to the ablation of two hydrogen atoms from succinate. Quinone (coenzyme Q) is reduced according to the equation:Q + 2*e*^−^ + 2H+ ➔ QH_2_

The atomic structure of SDH is known from crystallographic studies. The pathway for electrons from succinate to quinone resides within SDH; in contrast, protons are expected to be released in/captured from the environment. SDH is an enzymatic complex with four subunits (SDHA-D). A simplified point of view (Figure 3) would consider that there is a succinate dehydrogenase (SDHA) that sends electrons to a membranous enzyme (SDHC-D), reducing the mitochondrial coenzyme Q (quinone) that is the lipophilic redox shuttle present in the mitochondrial inner membrane (Figure 2). Hence, although SDH/complex II is a single entity, it might be pertinent to sometimes consider separately the SDH activity and injection of electrons in the MRC (Figure 3 bottom). This is reflected by the history of SDH, because initial preparation yielded a soluble SDH with two subunits that could bind substrates or inhibitors [13]. This soluble enzyme contained FAD and iron; it could reduce several acceptors such as phenazine methosulfate (PMS) or ferricyanide [14], but did not interact with the respiratory chain, nor could it reduce quinone, and consequently complex II was considered to be distinct from SDH. Association of the soluble form of SDH with preparations derived from the mitochondrial inner membrane led to physical association between SDH and complex II activities [15]. Consequently, SDH and complex II were recognized as the same entity. Carboxin (an SDHI, Figure 1) was instrumental in these studies because it inhibited the complete enzyme (quinone reduction) but not the soluble SDH.

### 3.1. Measurement of SDH Activity

Measurement of the SDH/complex II activity uses a terminal electron acceptor such as 2,6-dichlorophenol-indophenol (DCPIP) for enzymatic measurements in solution [16] or tetrazolium salt for histochemistry [6] (Figure 4). Their reduction leads to the formation of a product quantitated by optical absorbance measurement. These enzymatic dosages also involve an intermediate redox shuttle such as decylubiquinone or PMS, the latter being dispensable. A criterion to discriminate SDH from other pathways able to reduce the acceptors is the use of malonate that inhibits the SDH reaction. The reduction of tetrazolium salts may occur with a reaction restricted to SDH, while a procedure using decylubiquinone and DCPIP probes for the whole physiological range of the reaction: SDH and complex II. Tetrazolium salts are used for the cell viability test, “MTT”. The difference relies on the nature of the electron donor. For SDH measurement it is succinate, and for the MTT test it is the cellular metabolism, which means several dehydrogenase activities and, therefore, includes but is not limited to SDH.

Another evaluation of SDH activity comes from experiments in which mitochondrial respiration is recorded in the presence of succinate as the sole substrate. In this case, oxygen consumption reflects succinate oxidation by SDH, but in addition it requires the transport of external succinate into the mitochondria, and complex III and IV activities. It may therefore underestimate the maximal SDH activity.

### 3.2. SDH Affinity for Succinate and Maximal Enzymatic Activity

The values of the affinity constant of SDH for succinate (Km) range from 0.5 to 3 mM examples in [6,17]. The biochemical dosage of the SDH activity requires a sustained reaction rate with SDH at its Vmax, and a concentration of 20 mM is recommended [16]. The maximal enzymatic activity of SDH is rather high. For example, the fastest oxygen consumption rate observed with isolated liver mitochondria is obtained by recruiting SDH activity with 5–20 mM succinate. Biochemical dosages of respiratory complexes in beef heart, human muscle, and human liver indicated activities of complex II equal (heart) or superior (muscle, liver) to that of complex I [16]. Within a single study, the ratios between maximal enzymatic activities of complexes I, II, and V were determined in four mouse tissues [18], and the ratios of complex II/complex V approached 0.5 in the four tissues. One reaction of complex II would feed 1.6 reactions of ADP phosphorylation by complex V [19], hence a maximal activity of complex II alone would approach 80% of saturation of complex V activity, although normal metabolic constraints imply much higher recruitment of complex I activity (see below). Similarly, the ratio of complex II/complex I activities was below 0.5 in the brain (where complex I is overrepresented), between 0.5 and 1 in muscle or heart, and between 1 and 1.5 in liver. Under standard physiological conditions SDH is therefore expected to operate in vivo at a rate considerably lower than its possible maximal velocity, and suggests succinate concentrations below its Km. Consequently, in most cases it is assumed that there exists a significant “reserve of SDH activity” that is not recruited.

### 3.3. Chemical Inhibitors of SDH

The duality of SDH/complex II is reflected by the known inhibitors of this enzymatic complex (Figure 5). Hydrophilic analogues of substrates such as malonate [20,21,22], nitropropionic acid (NPA) [23], or oxaloacetate are small, water soluble, organic acids that inhibit the “SDH side,” actually SDHA [24]. In the plants where it is present, NPA is an intermediate in nitrogen metabolism, and is also considered a defense against herbivores. Oxaloacetate and malate are intermediates of cellular metabolism, and inhibition of SDH by oxaloacetate [25], and possibly indirectly by malate [26], appears physiologically relevant, both resulting from oxidation of fumarate, the product of SDH activity. Altogether, SDH in vivo appears to be stimulated by the concentration of its substrate succinate and repressed by metabolites derived from its product.

TTFA, atpenin, and pesticides known as “SDHIs” are molecules that disturb the electron transfer to quinone, and are consequently expected to act downstream of SDHA and close to the complex II side [24]. The structure of different SDHIs is shown (Figure 5), and they share common properties with regard to molecular weight, structures, and physical properties. On the grounds of the few data available and the computation of estimations Log P (Figure 1), the solubility of SDHIs appears to be far greater in organic solvent or lipids than in water. This hydrophobic character could influence the apparent affinity between inhibitor and target. A strongly hydrophobic inhibitor would accumulate in the membrane and reach disproportionately higher concentrations than that expressed in mg/kg or in moles/L, taking into account a diffusion space that is essentially aqueous. Then, a membrane protein such as SDH could be saturated by a hydrophobic poison concentrated in the lipidic phase, and this even if the tightness of binding (affinity of the protein binding site) is low. The biochemical preparation of a membranous protein such as SDH implies the presence of lipids and detergent forming a micellar structure bound to the hydrophobic domains of the protein. This means a strongly reduced volume of lipidic phase in contrast with the native situation of the cell/mitochondria. This may explain the contradiction between the high potency of carboxin to inhibit preparation of SDH and, upon dilution, a full reversibility of carboxin binding to the enzyme [2]. In contrast, if the molecule is hydrophobic and, furthermore, shows high affinity for its protein target, as with mitochondrial inhibitors such as rotenone, oligomycin, or antimycin, the release from the protein binding site is extremely slow and inhibition becomes essentially irreversible. Consequently, even if levels of exposure are low, binding to the target becomes cumulative and inhibition aggravates with time. Experienced users are aware of contamination of experiments/materials with hydrophobic high affinity inhibitors. The extent to which SDH inhibition by SDHIs is reversible appears, therefore, an issue with regard to their toxicity.

### 3.4. Reactive Oxygen Species Production by SDH

The electron path in the SDH complex starts from the SDH reaction in SDHA and includes several steps before reaching the acceptor quinone (Figure 4). During its reaction cycle, SDH is therefore in an intermediate redox state, hence partially reduced/oxidized depending on the relative speeds of these successive electron transfer steps. Electrons could escape to this normal path and be captured by other acceptors. Single electron transfer to oxygen would generate superoxide, and the transfer of two electrons hydrogen peroxide (H_2_O_2_), both of which are reactive oxygen species (ROS). This mechanism for ROS is predicted to be affected in an opposite manner by the different inhibitors: substrate analogs (malonate, NPA) would prevent electron entry into SDH, and hence leave no opportunity for leakage of electrons from SDH to oxygen. In contrast, SDHIs impede exit to quinone, would cause a blockade of electrons within the enzyme, and would promote the leakage of oxygen, hence ROS production. SDH can also be causative of mitochondrial ROS release by an indirect mechanism related to redox interactions with other components of the respiratory chain (see Section 6.3).

### 3.5. Role of SDH in the Tricarboxylic Cycle

The tricarboxylic acid cycle (TCA) is the major pathway to oxidize carbon atoms contained in energetic substrates (carbohydrates, fatty acids, amino acids) into carbon dioxide (Figure 6a). SDH reaction takes place after oxidation of the two carbon atoms and initiates the oxidation of succinate to regenerate the acceptor oxaloacetate, allowing a subsequent cycle of oxidation. The products of the TCA relevant for cellular energy supply are one molecule of ATP (Figure 6) and four pairs of electrons, three stored on the intermediate NADH and one generated by the SDH reaction.

It makes a big difference if one considers the SDH reaction from the point of view of electron transfer from complex II in the respiratory chain, or from the point of view of SDH participation to the TCA oxidation pathway (Figure 7). The complex II reaction yields two electrons to the MRC, and the transfer of these electrons to oxygen results in 1.6 ATP [19]. In contrast, the SDH reaction is mandatory for a complete TCA cycle and therefore drives the generation of three NADH equivalents to 3 × 2.7 = 8.1 ATP [19], plus one from succinyl CoA hydrolysis, hence 9.1 ATP.

If complete oxidation of glucose is considered, the theoretical yield is 32–34 ATPs. It implies two rounds of TCA, hence two complex II reactions contributing for 3.2 ATP, approximately 10%. In other words, if the activity of SDH and of complex II would be considered separately, invalidation of SDH would have much more severe consequences for cellular bioenergetics than that of complex II. The question is, then, whether this finds a reflection in different influences for inhibitors of SDH (SDHA): malonate, NPA, and of inhibitors of complex II (SDHC, SDHD), such as SDHIs impacting on electron transfer to quinone. Notably, in a study evaluating the toxicity of the SDHI fluxapyroxad in zebrafish [27], the authors used two different methods for measurement of the enzymatic activity of the SDHI target; one was supposed to measure the activity of SDH, and the other complex II activity. In the presence of the SDHI fluxapyroxad, the activity of SDH appeared increased and, at the opposite, that of complex II decreased. It suggests that the activity and/or expression of the SDH might be dissociated from that of the whole SDH/complex II entity, and therefore may reflect in vivo the lability of association in the whole enzymatic complex.

The scheme of the TCA used before with release of two CO_2_ and cyclic renewal of oxaoacetate (Figure 6a) reflects full oxidation of glucose and of fatty acids, but not, for example, oxidation of amino acids (Figure 8). TCA is also a hub for biosynthesis starting from six, five, of four carbon intermediates; these exits of one TCA intermediate are to be compensated by entries, hence TCA does not operate in a cyclic mode, but linear sets of reactions are to be considered (Figure 9b). If biosynthetic intermediates are collected upstream from SDH, inhibition of the latter would not be an obstacle, and indirectly may rather promote diversion of these intermediates toward biosynthesis. In addition, while the first part from formation of citrate to succinylCoA is driven clockwise by three reactions with strong negative ΔGs, the second part from succinate to oxaloacetate is much more reversible (Figure 6b). Notably, it includes a NAD/NADH redox step by malate dehydrogenase, the last reaction of TCA, and then, in addition to the quinone reduction state, the NAD/NADH ratio influences the SDH reaction, making it sensitive to the redox state of both the respiratory chain and metabolism.

## 4. Loss of SDH Activity

Genetic deletion of SDHD in mice results in loss of SDH activity, and is embryonically lethal [7]. This same study showed the limited influence of the loss of one allele of the SDHD gene in heterozygote mice, although the SDH activity is reduced by half. In humans, severe loss of SDH activity causes Leigh syndrome, which is also caused by mutations in the other mitochondrial complexes I, IV, and V. This indicates that common traits result from impairment of the mitochondrial Oxphos. Among these are lesions in the neural system, and notably the brainstem and basal ganglia, and there are privileged targets of mutations affecting Oxphos, although the mutation is present everywhere. Of course, as the severity of mutation increases, it affects most vital functions. Lactic acid increase in blood is a shared consequence between “mitochondrial diseases” because lactic fermentation aims to substitute the deficient Oxphos.

### 4.1. SDH Negative Tumors

Mutations in SDH were found in various tumors [28]. These tumors are hereditary paraganglioma or hereditary pheochromocytoma. Loss of SDHB expression is observed and actually used as a diagnostic criterion [29]. Tumor cells contrast with neighboring (non-tumoral) SDHB positive cells. This indicates that tumor formation is associated with the loss of the valid SDH allele in a heterozygous individual. However, not all of these tumors are hereditary, and other genes might be involved, but still SDHB could be affected while other subunits of SDH appear protected from degradation [28,30]. Unfortunately, the enzymatic activity of the immunoreactive intracellular SDHA was not documented.

### 4.2. Inhibition of SDH in Experimental Models

Biochemists use malonate (competitive inhibitor) to tune down the SDH activity in vitro or in vivo. In animal studies, nitropropionic acid (NPA) is used to irreversibly inactivate SDH, and to trigger striatal lesions recalling those observed in Huntington’s disease. A 30 to 60% inhibition of SDH is required for the observation of lesions [6]. This is fully consistent with the genetic data in mice and in humans with pathogenic mutations in SDH that are recessive. Notably, at lower doses, SDH inhibitors malonate [31] or NPA [10] induce a better resistance to a subsequent incident. This paradoxical effect is observed with many other events adverse to mitochondrial respiration (such as short duration hypoxia), and is called preconditioning. It is thought to result from the induction of adaptive/protective mechanisms. A consequence is that the relationship between dosage and effect changes its nature when considered over a wide range, and in experiments with no predictions about the possible effect(s), it may lead to disregarding mitochondrial inhibition as causative for observations that become inconsistent from one dosage to another.

## 5. Consequences of SDH Inhibition

The deleterious effect of genetic deficiency in SDH or of SDH inhibitors (3-NP, malonate) are thought to result from energy metabolism impairment. With regard to inhibition, three parameters would condition the consequences of the administration of an inhibitor:Is inhibition complete or partial?If inhibition is partial, could the remaining enzyme activity be sufficient to ensure normal reaction rate?Are there salvage pathways that would compensate for or alleviate the consequences of inhibition of the target enzyme?

A full blockade of SDH would result in the fact that TCA may proceed up to succinate, then two NADH and one ATP per succinate formed would be obtained, representing 6.4 ATPs. However, in this biochemical scheme (Figure 9c), oxaloacetate becomes a substrate that will be exhausted, preventing further introduction of acetyl groups in the oxidation pathway. The production of ATP and NADH by the TCA would stop, oxidative metabolism and Oxphos would stall, and this would result in cell death within a short time.

### 5.1. Salvage Pathways to SDH Inhibition

#### 5.1.1. Succinate Accumulation

A salvage pathway is conceivable if oxaloacetate is generated continuously by other means than succinate oxidation. If so, the second problem is tolerance to the release/accumulation of succinate. To understand the formation of oxaloacetate from other sources than oxidation of succinate, the TCA cycle must be considered in a wider sense, with connections to other biochemical pathways (Figure 8).

Accordingly, oxaloacetate formation is possible from pyruvate (Figure 8 reaction 3) or from several amino acids (Figure 8: amino acid entries “c, d,” or “a” via pyruvate). The consequences of the engagement of this salvage pathway could be envisaged on the grounds of simple models (Appendix A). If oxaloacetate supply is guaranteed, the loss in SDH activity decreases moderately the yield in ATP for each round of TCA and increases the yield in ATP per oxygen. Hence, in terms of cellular bioenergetics, the absence/inhibition of SDH appears tolerable if the two metabolic challenges it creates are resolved. The first challenge is whether or not the cellular metabolism could be reshaped, and if glucose or amino acid supply would be sufficient to ensure continuous regeneration of oxaloacetate. This highlights the importance of metabolic reserves that would make germinating spores of fungi particularly sensitive to SDHIs when compared to a larger/more complex organism. The second challenge is the tolerance to/fate of the succinate accumulated.

#### 5.1.2. Succinate Cycle

Invalidation of the SDHD gene in mice showed that no adaptive mechanism is able to compensate for a complete loss of SDH. However, the occurrence of SDH negative tumors highlights that “SDH-KO” cells remain viable in a generally SDH positive individual. One possibility would be that the succinate generated by a truncated carbon oxidation, as shown above (Figure 9c), is used by other cells in the same individual. This would establish a succinate cycle, in which SDH deficient cells behave as succinate producers while others are succinate consumers. This proposal replicates what is known for lactate, although differences can be anticipated, and tallies with existing knowledge in adaptive/comparative physiology (see Section 6.3). In addition, it fits with the conditions for human exposure to SDHIs that are expected to be heterogenous and dependent on the entry route: inhalation, skin, or oral absorption.

#### 5.1.3. Electron Escape from SDHA

Electron escape from respiratory complexes (I, II, III) is essentially a consequence of the complexity of electron pathways within respiratory complexes, and is the explanation for reactive oxygen species (ROS) formation by MRC. Accordingly, a possible salvage pathway to complex II mutation or inhibition by SDHIs would exist if electron escape from SDHA became sufficient to trigger a TCA cycle activity and ATP formation compatible with cellular viability, but at the cost of enhanced ROS release (Figure 10). The rate depends on the efficiency of electron transfer [32]. Whether the response to fluxapyroxad in the zebrafish (see Section 3.5), as well as the loss of SDHB in tumors, reflects this adaptive response remains to be determined. Catalase renders cells quite tolerant to intense H_2_O_2_ exposure in the short term [33], and would neutralize the H_2_O_2_ flux generated by this modified SDH reaction. This salvage pathway is obviously impossible with inhibitors that target the SDHA, such as NPA or malonate. This “SDHA only reaction” would remain undetectable if the test relies on the presence of the complete SDH/complex II reaction—oxidation of succinate and reduction of the acceptor quinone—but would be included when other acceptors are considered (Figure 4 and Section 3.5).

### 5.2. Kinetic Compensation by SDH

An examination of the possible salvage pathways could explain why certain cells could survive to SDH ablation. This implies a rearrangement of metabolic fluxes. However, this rearrangement appears unnecessary if the inhibition of SDH is moderate. As explained before (see Section 3.2), present data suggests that SDH activity is largely excessive with regard to the flux of SDH reactions necessary for normal (glucose, fatty acids) oxidative metabolism, and, therefore SDH activity is likely to be limited by succinate concentrations in the range at or below its affinity constant. In these conditions a partial inhibition of SDH would increase the succinate concentration, stimulating SDH activity with restoration of a normal SDH flux, and therefore oxidation fluxes would remain unchanged (Appendix A).

### 5.3. Adaptive Responses to SDH Inhibition

The adaptive responses to increasing inhibition of SDH are summarized below with their consequences (Figure 11). Therefore, in the hypothesis of a limited exposure to contaminating SDHIs, it appears likely that the kinetic properties of the mammalian SDH would be able to compensate for inhibition and maintain a normal SDH reaction rate, with no other consequence than an increase in the steady state concentration of succinate. This may explain the tolerance of mammals to these inhibitors.

However, this analysis would not be complete without considering other facts that may reveal possible risks of SDHI exposure:Full aerobic oxidation is not always/everywhere pertinent and we should consider other metabolic schemes.Is a steady state increase in succinate concentration without consequences?The interaction between SDHIs and other active substances is to be considered.Finally, are the experimental conditions of exposure to SDHIs appropriate to evaluate their possible impact, and particularly with regard to differences between animal models and humans?

## 6. Complex II and the Hypoxic Challenge

### 6.1. The Hypoxic Challenge

A large number of evidences indicates that oxygen supply is the critical issue with regard to cellular bioenergetics. Oxygen or glucose supply from blood is roughly equivalent (3–5 mM), however delivery of oxygen to tissues is dependent on the dissociation of hemoglobin, and measurement of oxygen concentration immediately outside the capillary indicates 16 mmHg or less [34]. Oxygen and glucose concentrations in the extracellular medium differ, therefore, by orders of magnitude. This contrasts with a more intense flux of oxygen than of substrates (complete oxidation of one glucose requires six O_2_). Oxygen supply to cells becomes, therefore, a most critical issue that limits the extent of hypermetabolic states (exercise, neural stimulation, inflammation, cancer) or when vasculature is deteriorated (clot, trauma, inflammation). When the imbalance between oxygen supply and need jeopardizes cellular fate, the only possible strategy for adaptation in the short term is to restrain cellular oxygen use. A critical issue is, therefore, the efficiency of oxygen to generate ATP, quantified by the ATP/O_2_ ratio. A limited increase of it could be obtained by a rearrangement of the oxidative metabolism, such as a shift from fatty acid to glucose oxidation or, notably, if SDH activity is repressed but recruitment of anaerobic ATP formation pathways is required for a significant improvement [35].

### 6.2. Lactic Fermentation and Oxygen Debt: Later or Elsewhere

The best-known example of an anaerobic ATP formation pathway is lactic fermentation, which does not need oxygen at all. Hence, its ATP/O_2_ is irrelevant (infinite), and a modest contribution to energy metabolism increases considerably the cellular ATP/O_2_; this issue might be overlooked when the Warburg effect is considered [35]. Therefore, as the contribution of lactic fermentation to ATP regeneration increases, ATP/O_2_ rises considerably. Accumulation of lactic acid is considered, then, as an “oxygen debt” rather than as a waste, because lactate is oxidized later, or elsewhere. Elsewhere means that a cellular symbiosis could be established on the grounds of a “lactate cycle” between lactate producers and lactate consumers. This takes place between red blood cells and liver, or glial cells and neurons. Consequently, lactic fermentation cannot be considered only as an emergency salvage pathway under hypoxia, but is an essential component of the normal physiology of the mammalian organism [36].

### 6.3. The Mitochondrial Respiratory Chain in Anaerobic Mode

Another anaerobic ATP production based on the cooperation between complex I and II was evidenced. In this process, complex II is working in reverse mode and the reduction of fumarate into succinate allows the reoxidation of the quinone reduced by complex I [12]. The four protons pumped by complex I are used by complex V to generate ATP. This is, therefore, an “anaerobic Oxphos” based on a short respiratory chain in which fumarate replaces oxygen as the final electron acceptor (Figure 12). This anaerobic Oxphos was considered in animals showing extended adaptation to hypoxic environments, including parasitic helminths [37,38,39]. However, in mammals it was suggested [40] and later on evidenced in ischemic conditions [12]. The reversion of enzymatic reactions is propagated to the TCA cycle in the four carbons sector to feed SDH with fumarate (Figure 6b and Figure 9f), with intense consumption of metabolic intermediates susceptible to providing fumarate, malate, oxaloacetate. When oxygen comes back, succinate oxidation takes place. It becomes predominant and contributes to reperfusion damages, because administration of SDH inhibitor malonate during reperfusion improved the outcome in experimental models for heart ischemia/reperfusion damages [12,31,41]. The mechanistic explanation would be that the succinate accumulated during ischemia greatly stimulates the SDH activity that causes intense quinone reduction and the formation of a large membrane potential. Both oppose the normal reaction of complex I and, furthermore, drive reverse electron transfer by complex I (the oxidation of quinone and reduction of NAD), a reaction greatly increasing ROS formation and, consequently, oxidative damage. The metabolic consequence is that further oxidation of fumarate/malate is impaired (Figure 9g), which appears adapted for a sustained redox cycle between anaerobic succinate producers and aerobic succinate oxidizers.

As for lactic fermentation/reoxidation, the question of the contribution of a succinate cycle to normal physiology deserves consideration, and in mammals the retina appears as a first example [42]. Accordingly, the retina exports succinate to the retinal pigment epithelium-choroid that oxidizes it with SDH and exports malate that can be used by the retina for regeneration of the acceptor fumarate. A comparison between lactate and succinate cycles is proposed (Appendix A).

### 6.4. High Requirement for Electron Flux in Complexes I and II under Anaerobic Mode

In sharp contrast, with normal oxidative metabolism, the reverse electron transfer in complex II should take place at the same rate as the forward reaction in complex I (Figure 11). In addition, because of the low yield (1.08 ATP per reaction), the dependence of ATP formation on the fluxes in complex I and II is different. If the complete oxidation of glucose is taken as a reference for the aerobic pathway, the figures are 30 ATP from Oxphos with 3.2 from two complex II reactions, and the rest from the 10 complex I reactions. For the same 30 ATPs, 30/1.08 ≈ 28 reactions, as shown in Figure 12a, are necessary. Hence, while complex I activity is roughly multiplied by three (10 to 28), that of complex II is to be increased by a factor 28/2 = 14 times. The intense activity required during this anaerobic Oxphos has the result that a low inhibition to complex II (with marginal influence on the aerobic Oxphos) becomes critical. The same applies to complex I. A defect in oxygen supply is not restricted to ischemic events and, for example, inflammation or trauma deteriorate oxygen supply to tissues. Neural activation is a strong driver of cellular hypermetabolism, and failure to comply with the large increase in energy demand underlies excitotoxicity [43]. If some cellular domains resort to this anaerobic Oxphos, they would be considerably more sensitive to complex I and complex II inhibitors, and would be damaged well before all the rest relying on normal aerobic respiration/metabolism.

### 6.5. Succinate Is a Signal

As explained before, SDH integrates redox status, and increases in the reduction state of quinone and NAD opposes the forward reaction, that may then be restored by the increase in succinate concentration it causes, the same mechanism as the kinetic compensation above. In addition, exclusion of succinate from oxidation or reversion of SDH, hence succinate accumulation, increases ATP/O_2_. Therefore, succinate concentration appears as a sensitive metabolic clue to report the redox imbalance that could originate from a deficit in oxygen supply, and this over the full range: from modest decrease in oxygen concentration (kinetic compensation by SDH) to full anaerobiosis (reversion of SDH). Blood succinate concentrations are in the micromolar range [44], hence considerably lower than that of lactate. However, an increase in blood succinate is indicative of life-threatening critical conditions [45], of higher cardiovascular risk [46], and degraded metabolic status [47,48]. Subsequent consequences of an elevation in succinate relate to succinate signaling pathways, including receptors, protein, and epigenetic modifications. They are out of the scope of the present document, and are not considered further here. Nevertheless, it should be borne in mind that the metabolic consequences of exposure to SDHIs would increase succinate signaling in the absence of genuine hypoxia.

## 7. Metabolic Events

To be relevant with regard to a prolongated chronic low exposure such as the contamination of an environment (food), the impact should result in modifications able to accumulate with time, up to the point they could initiate a pathogenic sequence.

### 7.1. Mild Inhibition of SDH and Lipid Synthesis

Linear sequences of reactions in the TCA with entry and exit are essential for biosynthesis of macromolecules; it is also of great importance for metabolic responses to feeding/starvation. In the fed state, an excess of calories is stored as fat, a significant part coming from the conversion of glucose (but also proteins) into fatty acids. The starting point is the citrate (Figure 9d) formed by condensation of acetylCoA (C_2_) and oxaloacetate (C_4_); TCA recruitment is minimal and an SDH reaction is not required (Figure 9d). Under starvation, a critical issue is the generation of glucose from other sources than glycogen that is quickly exhausted. The major source is proteins broken down in amino acids, which, after the loss of their heteroatoms (N, S), are converted into molecules that could be injected into the TCA (Figure 9e) with the purpose of generating C_4_ molecules malate and oxaloacetate, the starting point for gluconeogenesis. Therefore, in contrast with lipogenesis, the recruitment of TCA reactions is intense and requires SDH for certain amino acids and, notably, glutamine (Figure 8). Kinetic compensation for partial inhibition of SDH (Figure 10) relies on steady state increase in succinate concentration that is proportionate to exposure to the SDHI. Consequences are expected to propagate over the entire TCA, with accumulation of the upstream metabolic intermediates, and a depletion in the downstream products. Upstream intermediates include citrate, the precursor of lipogenesis, and downstream intermediates include precursors of gluconeogenesis (malate oxaloacetate). SDH inhibition would therefore result in a higher tendency to accumulate fat, while gluconeogenesis (adaptation to starvation) would be impaired. In other words, to preserve normal energy metabolism (ATP turnover rate), animals tend to accumulate more fat. Accordingly, when animals are exposed to increasing dosages of SDHI, an observation of weight gain before adverse effects appear with higher doses should raise concerns about this metabolic issue.

### 7.2. Highest Exposure to Oral SDHI Gut and Liver

When oral administration/exposure is considered, the organs most exposed to SDHIs are the gastrointestinal tract (GI) and the liver (Figure 13). The GI includes microbiota that also express SDH, and would be first target for ingested SDHIs. The GI itself faces rather adverse conditions with regard to mitochondrial respiration as, for example, microbiota release hydrogen sulfide, a poison for cellular respiration comparable to cyanide. Adaptations to this deeply modify mitochondrial metabolism, with impacts on the relative role of respiratory complexes, hence SDH [49,50] and complex I [51]. Consequences of exposure to SDHIs in the GI are, therefore, expected to be of a different nature than in the rest of the mammalian organism. The liver actively metabolizes SDHIs to release degradation products eliminated in the bile. Hepatic cells metabolizing SDHIs are, therefore, in one of the states of SDHI inhibition shown above. Human exposure, within acceptable limits, is expected to result in a low level of inhibition. State 2 (Figure 10) might be considered, but remains uncertain. In contrast, state 1 is a direct consequence of the intracellular metabolism of SDHIs, with the indirect consequence of increasing the flux of lipogenesis, which would result in a cumulative effect leading to lipid accumulation in the liver, increasing the risk of Non-Alcoholic Fatty-Liver Disease (NAFLD).

This NAFLD is a disease in rapid progression, and because of its consequences—insulin resistance and Non-Alcoholic Steato-Hepatitis (NASH, with increased risk of liver cancer)—it is now the first cause for liver transplantation. This effect was likely to escape experimental studies for several reasons:Its cumulative nature needs a long time to develop.It appears unlikely that inquiries about effects related to metabolic diseases were required in the process of evaluation of these active substances.Whenever hepatic carcinogenesis is recorded in rodents, it might be considered as specific to rodents and irrelevant to human exposure. In this respect, comparison with transgenic animals that, equipped with SDH, mutated to acquire resistance to a given SDHI would help to dissociate effects caused by exposure to the xenobiotic (SDHI) from that resulting from SDH inhibition.Unless the effect would be extremely potent, it would not develop spontaneously, but would require diet manipulation (high fat, high sucrose diet), and rodents are relatively resistant to it.

### 7.3. Metabolic Issues with Animal Models

Rodents are a poor surrogate for metabolic studies. A first issue is their hypermetabolic nature, because the contribution of adaptive thermogenesis to metabolic rate is considerable in small homeotherms and may be subject to variations with little/no detectable consequences, unless a careful examination of behavior and quantitation of energy expenditure would be made. The impact of thermogenesis on energy expenditure could be estimated from the difference between basal metabolic rate (BMR), including, therefore, thermogenesis with daily energy expenditure of hibernation (DEEH) supposed to represent the preservation of vital functions in the absence of thermogenesis. The difference between BMR > DEEH decreases with size (body weight), and is predicted to be null for a body weight subject to contradictory evaluations [52,53]. This may severely downgrade the gain in energy expenditure brought by hibernation in man [54]. A second issue is that establishment of NAFLD/NASH in mice by diet is subject to uncontrolled variations even if protocols are known for long time. Uncontrolled experimental conditions such as variation of micronutrients in the diet may explain this. Adverse effects linked to the genetics of laboratory strains might also contribute to their resistance to metabolic diseases. There is evidence that the alleles of genes implicated nowadays in metabolic diseases are related to those that ensured a “high metabolic efficiency,” hence survival when food was scarce: the “thrifty gene” theory [55,56,57]. This is thought to explain the high prevalence of metabolic disorders at present in human groups exposed for several generations to severely adverse food supply conditions in the past. The best-known example is the Pima Indians [58,59,60,61]. Most experiments with animal models have resorted to homozygous strains. The establishment of a homozygous strain implies a strong selection of unique alleles for all essential genes. This takes place under environmental conditions for which a high metabolic efficiency is of no selective value, since animals were usually fed ad libitum. In practical terms, the impact of a given molecule on a metabolic syndrome would require animal models alternative to homozygous strains of laboratory rats and mice [62]. In addition to food excess and deficit of exercise, a chronic inflammation state is thought to aggravate metabolic status. While other stress factors, such as immune challenge, are more difficult to evaluate, one may also suspect that they were limited during the establishment of homozygous strains in comparison to those imposed in “normal life.” Finally, experimental protocols imply animals free of immune challenge and with a supposed minimal stress level.

Zebrafish proved to be highly sensitive to SDHIs [63]; this is true also for xenopus in which the synergy between SDHIs and strobilurins (inhibitors of complex III) was demonstrated [64]. Both probably reflect the high sensitivity of aquatic life to mitochondrial poisons, a sensitivity widely used for fishing or control of populations [65,66,67]. The last authorized use of rotenone relates to its piscicide activity [4]. Several factors could explain this high sensitivity of aquatic organisms: (i) Their body temperature is imposed by the environment, not thermogenic activity. Hence, most metabolic activity is used for vital functions (somehow, they are permanently under a regime close to DEEH), and therefore the threshold for tolerable relative inhibition of a metabolic enzyme is likely to be much lower than in the highly hypermetabolic rodents. (ii) Most of the time, exposure takes place by dilution of the toxic agent in water. Exposure is then amplified by the fact that extraction of the same amount of oxygen in water or air implies much larger volumes of water. (iii) The solubility of SDHIs (and other mitochondrial poisons) makes the cellular membranes privileged domain in comparison with the surrounding water, and where concentration would be greatly increased. For these two reasons, the contact between aquatic animals and SDHI diluted in water is expected to be much more intense than when terrestrial organisms are considered.

Finally, the influence of SDHIs on NAFLD, metabolic syndrome, obesity, or diabetes have been up to now poorly explored (Appendix A).

## 8. Convergent Interferences

### 8.1. Quinone

Studies on the mitochondrial respiratory chain pointed to quinone (Q in figures) as a convergence site for electrons provided by several enzymes (complex I, complex II, electron transfer flavoprotein, L-alphaglycerophosphate dehydrogenase, sulfide quinone (oxido)reductase, …). The shared quinone electron acceptor creates an indirect redox communication between these enzymes, and impacting on one would impact on the reaction rates for the others. Firstly, they could enter into competition to reduce quinone [25], with the inversion of reactions as the ultimate consequence documented for complex II [50] and complex I [12,51]. Secondly, their quinone sites could share some common properties and inhibitors that act as quinone analogs and may target more than one enzyme. Competition between inhibitors and quinone makes the quinone content of the mitochondrial inner membrane a critical factor, and the interaction of drugs thought to impact on quinone content should be considered. This is the case for the widely used statins. From quinone to oxygen electrons travelling through a single path, complex III or IV and inhibition of one or the other impact oxidative metabolism as a whole. Synergy between SDHI and complex III inhibitors was evidenced [64].

Interaction between quinone and these redox enzymes takes place in a constrained and probably confined lipidic environment. This issue is linked to the hypothesis of association of respiratory complexes into supercomplexes forming a “respirasome” in the mitochondrial inner membrane. In this respect, relationships between complexes I and II attracted attention [25]. Perturbation brought by any hydrophobic inhibitor could originate from two different types of interaction: (i) with the proteins, by affinity for the structural domain of the respiratory complex, (ii) with the lipids creating disturbances in the confined lipidic environment. The first should show specificity towards one of the complexes and be sensitive to mutations, which could cause resistance to the inhibitor. In contrast, the lipidic interference would be less specific, hardly affected by mutations, but: (i) sensitive to the lipidic environment, which may target a class of organisms; and (ii) susceptible to cumulative effects. For example, several different “lipidic troublemakers” could synergize to disturb the respiration in absence of, or in addition to, specific inhibition depending on their respective nature.

### 8.2. Mitochondrial Respiration

Targeting the energy metabolism of a competitor in the environment is a widely shared strategy; the most used agents for mitochondrial studies (rotenone, antimycin, oligomycin, …) are antibiotics acting on respiratory complexes, and this includes the use of SDHIs. A few pathogens target the mitochondria of their host. Selection of the Oxphos target would reflect its critical role in fitness. However, the mitochondrial oxidative metabolism is a frequent unwanted target of pharmacological agents, hence not intentionally selected on these grounds. An explanation might be the large number of metabolic steps, and the complexity of Oxphos, that make them sensitive to a large number of troublemakers. Oxphos relies heavily on a membranous domain that could increase their concentration, and may become itself a target (ionophores). In addition, mitochondrial membrane potential increases greatly the intramitochondrial concentration of permeant cations. Therefore, oxidative metabolism and notably Oxphos appear as a likely target for the summation of the marginal effects of different active molecules’ “cocktail effect,” with adverse consequences on the metabolic status that starts with the low levels increasing risk for pathologies but not pathogenic per se. This would require specific experimental [64] or theoretical approaches aiming to measure and evaluate the consequences of this heterogenous chronic exposure of populations.

As explained before, mild inhibition is usually resolved by a kinetic response that restores normal metabolic flux. When applied to mitochondrial respiration as a whole, the kinetic response would rely on the TCA cycle and other metabolic redox reactions, with an increased reduction state (NADH/NAD ratio) and an increased concentration of TCA intermediates. In other words, this kinetic compensation would preserve cellular energy metabolism at the cost of the increased availability of reductive power (electron overload) and precursors. This would stimulate electron leakage (ROS formation), lipid synthesis, promote metabolic syndromes and prime/reproduce the alterations associated with cancer metabolism. Many factors, and notably lifestyle, could contribute to this effect, but SDHIs are to be considered as possible contributors to it.

## 9. Assimilation of SDHIs and Diet

When the oral route is considered, two factors are expected to influence the final outcome: the toxicity of the active substance, and the extent to which it is internalized, meaning here how much it goes through the walls of the gastrointestinal tract. In this respect, the integrity of the gut barrier would be an issue. SDHIs are poorly soluble in water and this is expected to impair fast diffusion and transfer through the gut wall. This is also the case for a number of pharmaceutical drugs, and in order to improve their absorption a strategy is summarized as: “*In the simplest sense, coadministration of drugs with lipids in lipid-based formulations (LBF) seeks to harness the advantages of endogenous lipid processing pathways to support drug absorption…*” [68]. Accordingly, lipids or detergents (such as Tween 80) were used to improve drug delivery.

A specific attention should then be applied to protocols used to administrate the SDHIs to animals. If present in a standard diet, a tolerance to SDHIs might result from innocuity of the substance or from impaired absorption. If SDHIs are provided by gavage, inclusion of lipids or detergents would improve absorption and follow-up of accumulation and metabolic degradation in organs. However, if so, these experiments would be inappropriate to estimate the internal exposure with SDHIs present in the diet. This would have little consequence for the determination of acceptable levels for oral exposure in humans if the absorption parameters would be identical between laboratory animals and humans, but this appears unlikely. Lipids are natural components of the diet, and the human diet (and particularly the western diet) is richer in lipids than that of laboratory rodents. Tween80 is known as polysorbate and is a food additive (E433) used to stabilize emulsions, and consequently is likely to be overrepresented in human diet compared to that of laboratory animals. Finally, alcohol is a solvent present in the human diet. A last comment should be made on the fact that the formulation of the pesticide is not purely active substance(s) (powder), but usually a liquid emulsion/suspension to be diluted for use, and the physicochemical principles grounding preparation of such preparations may overlap somehow with those controlling oral assimilation mentioned before.

## 10. Conclusions

This report could certainly be criticized on the grounds of its theoretical and simplified nature. More substantial data from the literature could/should have been brought here to support the proposals made in this document. Firstly, this would mean a wide range of coverage with bioenergetics, preconditioning, metabolic control, specific mechanisms in different tissues, etc., out of range during preparation of this manuscript. The reader stimulated by this manuscript would probably easily find a path to improve accuracy in a given domain. Secondly, setting aside these obvious imperfections, this analysis allows:

### 10.1. To Understand How Cells/Organisms Withstand SDH Inhibition

There is a significant “reserve” in SDH activity that could be mobilized under conditions of partial inhibition. This results in a threshold effect and, for example, existing animal models exposed to the SDH inhibitor NPA require a 30% to 50% inhibition of the SDH activity to result into detectable lesions.In contrast with NPA that inhibits the metabolic side of SDH (SDH-A), SDHIs target the electron transfer from complex II (SDHB-D) to quinone, which is a minor contributor to MRC bioenergetics and mitochondrial ATP production.Heterogenous exposure to SDHIs resulting from the intake route, and degradation/elimination, would allow the compensation of inhibition of succinate oxidation in cells more exposed to SDHIs by a higher rate of oxidation in others.

### 10.2. To Identify

Possible risks associated with these adaptive mechanisms: the response to partial SDH inhibition induces moderate and proportionate metabolic alterations, which may have insidious effects when present over the long term (chronic exposure). Altogether, if a marginal exposure to SDHIs is to be considered the main risk, it appears their role is as one of the many environmental factors adverse to the oxidative metabolism (sometimes called “mitochondrial fitness”), contributing to metabolic overload and aggravation of metabolic status. With oral exposure, a liver with risk of NAFLD appears to be the first target.Consequently, for SDHIs and other molecules interfering with mitochondrial function, metabolic disorders might well become an issue. Investigations may require specific adaptations of animal models.Clearly identified a lack of knowledge: (i) Interference of SDHI with metabolic status is hardly documented (Appendix A). (ii) More extended studies of inhibition properties of SDHIs with regard to non-target species, and notably man, should be made public. A critical issue is the reversibility of the binding of SDHIs on SDH. (iii) The SDHI assimilation should be better evaluated; this includes the influence of diet as well as consideration of the formulations actually in use.Neglected metabolic schemes: while normal bioenergetics would be relatively indifferent to partial inhibition of SDH, tissue specific adaptations or events could greatly enhance sensitivity to SDH inhibition. In this respect, evaluation of active substances could hardly be considered satisfying if their interaction with life challenges, such as immune challenge, inflammation, and starvation are omitted.

### 10.3. To Determine Phenotypic Alterations That should Attract Attention

Greater vulnerability to NAFLD/NASH and metabolic syndrome:Impaired gluconeogenesis and adaptation to starvation,Body weight gain (lipid accumulation) likely to be biphasic,Modification in brain structures (brainstem and basal ganglia),Retinopathy,Failure to comply with an energetic challenge: intense repeated neuronal stimulation, high oxygen demanding organs, oxygen limitation,Alteration in hypoxic signaling and response to ischemic reperfusion events, and,Increased index of oxidative stress and/or induction of antioxidant defenses.

## Figures and Tables

**Figure 1 ijms-24-04045-f001:**
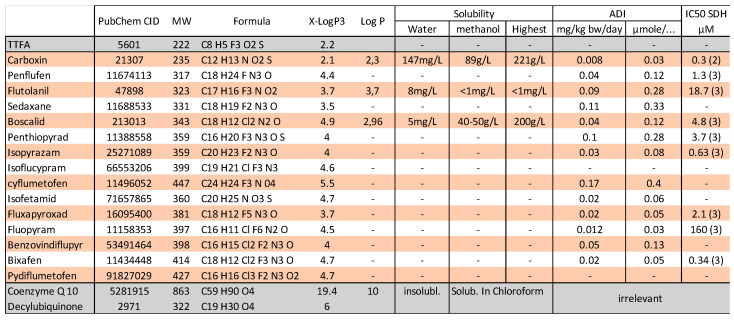
Data were obtained for the largest part from PubChem (https://pubchem.ncbi.nlm.nih.gov/, accessed on 19 December 2022). TTFA, decylubiquinone, and coenzyme Q10 (ubiquinone) are not SDHIs, but are relevant here (gray lines). The other lines (orange/white) list SDHIs presently in use. X-LogP3 is the predicted LogP value obtained by computation. The experimental LogP/LogKow (octanol/water) values and solubility data are indicated when available. ADI (acceptable Daily Intake) indicates the level of the limit for exposure in the general population. The last column shows the IC50 available (2) and (3) refer, respectively, to references [2,3] from which these values were derived.

**Figure 2 ijms-24-04045-f002:**
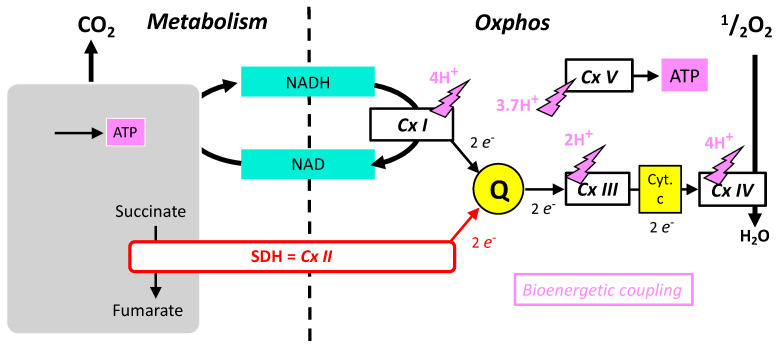
Schematic representation of the role of SDH and its relationships with oxidative metabolism (Metabolism), shown as a grey box releasing CO_2_ and producing some ATP and oxidative phosphorylation with the five complexes (CxI-V) and redox shuttles (NAD/NADH) quinone (Q) and cytochrome-c.

**Figure 3 ijms-24-04045-f003:**
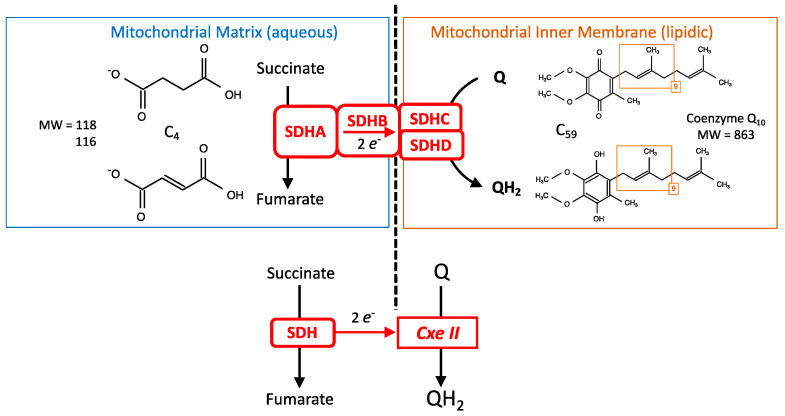
SDH (**Top**): Schematic representation of SDH subunits with the formulas of the substrate (succinate) and product (fumarate), as well as that of the electron acceptor, the oxidized Coenzyme Q_10_. (**Bottom**): Functional scheme with an “SDH side” and a “complex II side”.

**Figure 4 ijms-24-04045-f004:**
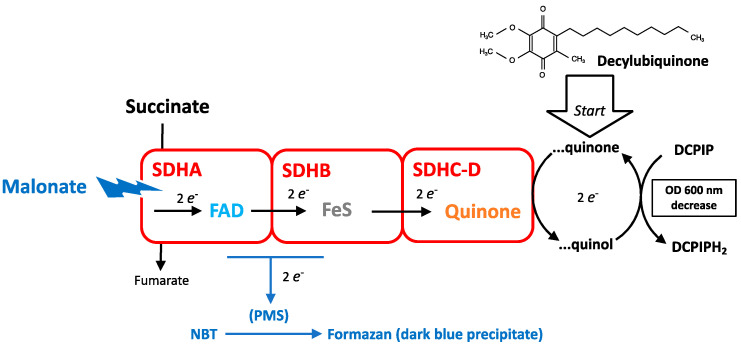
Measurement of SDH activity. Subunits A, B, and C-D are schematized by blocks; their most relevant redox cofactors are indicated: FAD, Iron sulfur centers (FeS), and quinone. Interception of electrons by acceptors used to assess SDH activity is shown. PMS is abbreviation for phenazin methosulfate, and NBT for nitro blue tetrazolium. With dichlorophenol-indophenol (DCPIP), the reaction is initiated by the addition of the redox intermediate decylubiquinone. Malonate, a competitive inhibitor for succinate, is used to check for the specificity of the reaction.

**Figure 5 ijms-24-04045-f005:**
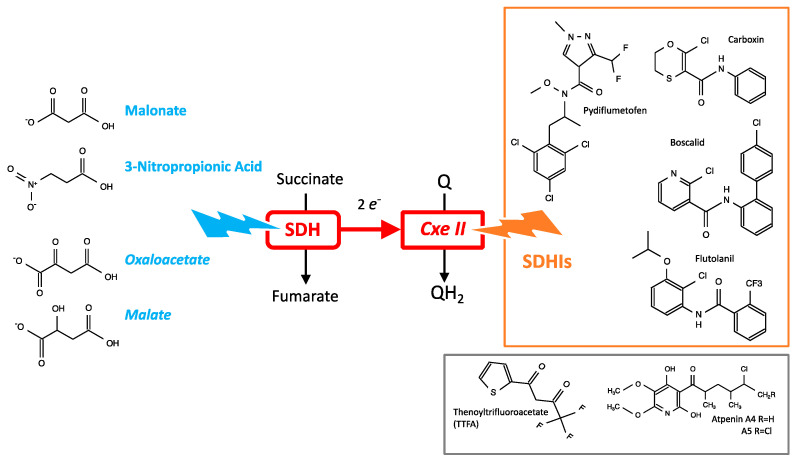
Inhibitors of SDH.

**Figure 6 ijms-24-04045-f006:**
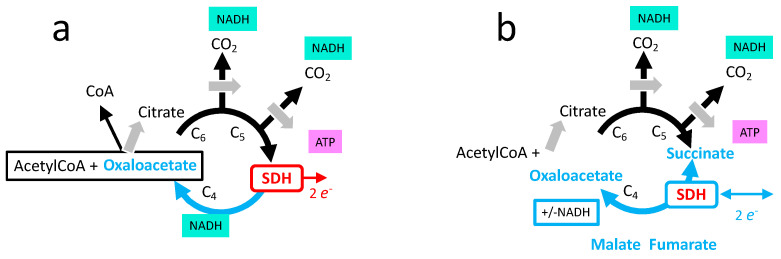
Tricarboxylic acid cycle (TCA). (**a**) Oxidative metabolism: one TCA cycle consumes the two carbons brought by condensation of acetylCoA with oxaloacetate. It ends with the regeneration of oxaloacetate. This implies three steps of NAD reduction into NADH, phosphorylation of one molecule of ADP/GDP into ATP/GTP (shown as ATP), and one reaction of the SDH. (**b**) Three reactions (gray arrows): citrate formation and the two release of CO_2_ steps have strong negative ΔGs and orient TCA clockwise (oxidation). In contrast, the sequence of reactions between succinate and oxaloacetate are reversible.

**Figure 7 ijms-24-04045-f007:**
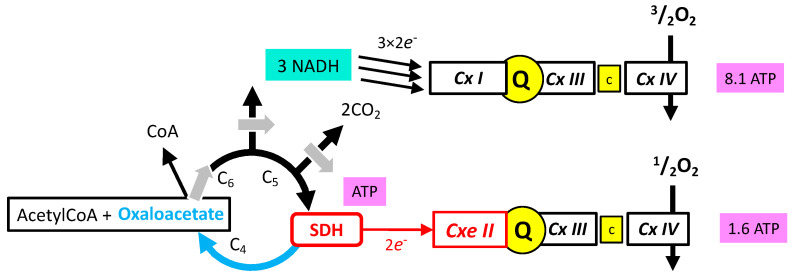
ATP formation resulting from one round of TCA. TCA itself generates one ATP. Entry into Oxphos by complex I (CxI) reoxidizes the 3 NADH (3 × 2.7 = 8.1 ATP), and by complex II (SDH) 1.6. These values [19] are the theoretical (maximal values) with full (100%) coupling between electron transfer (complexes I, III, IV or II, III, IV) and phosphorylation by complex V (not shown here). Depending on the authors, the physiological coupling ranges from almost 100% to 50%, and ATP yield for Oxphos would change accordingly. The production of ATP by TCA is stoichiometric with the reaction succinylCoA to succinate + acetylCoA, hence not affected by a coupling ratio.

**Figure 8 ijms-24-04045-f008:**
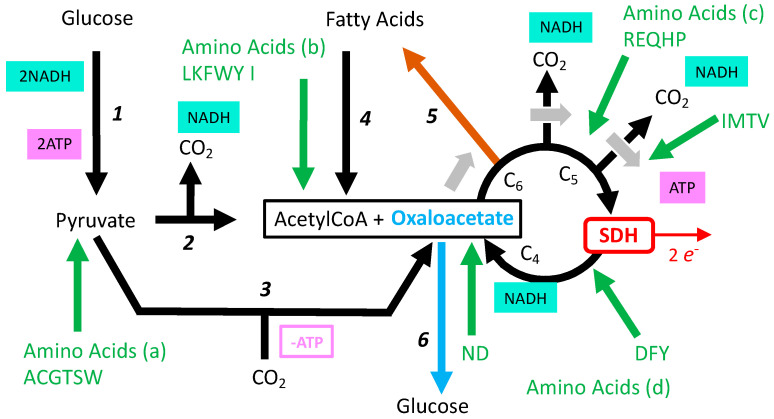
Interconnection of TCA with the rest of oxidative metabolism. 1: Glycolysis, 2: pyruvate dehydrogenase, 3: pyruvate carboxylase, 4: fatty acid oxidation, 5: fatty acid synthesis (lipogenesis), 6: gluconeogenesis, green arrows: different entries for the oxidation of the carbon skeleton of amino acids are shown with positions as relevant here (a–d), green capital letters refer to one letter amino acid code.

**Figure 9 ijms-24-04045-f009:**
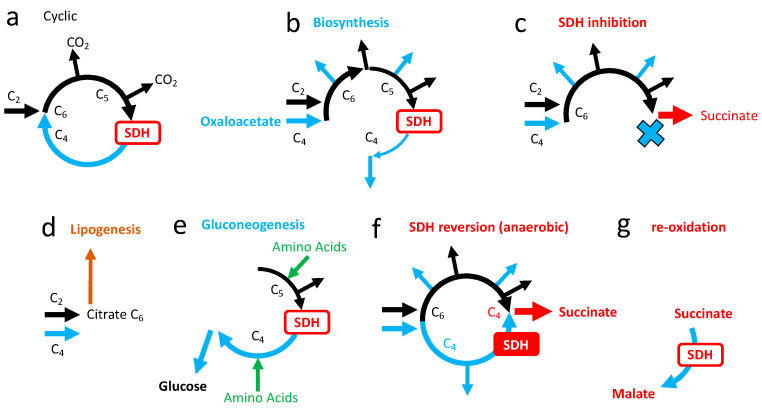
Simplified representation of different modes for the TCA cycle relevant here. Only the fate of carbons is taken into consideration (no representation for redox or phosphorylation reactions). (**a**): Cyclic oxidation of acetylCoA, (**b**): diversion of biosynthetic intermediates in C_6_, C_5_, or C_4_ (divergent blue arrows); these exits are to be compensated by refilling TCA with intermediates (anaplerosis), shown here as the introduction of a new oxaloacetate (horizontal blue arrow), (**c**): the inhibition of SDH succinate is the end product, (**d**): lipogenesis, (**e**): gluconeogenesis, (**f**): TCA in hypoxic/anaerobic conditions; succinate would be generated by two convergent mechanisms: (i) carbon oxidation from citrate to succinate, with improved ATP/O_2_ and moderate loss in metabolic efficiency, and (ii) by anaerobic Oxphos, implying reversion of SDH, (**g**): fast and exclusive oxidation of succinate to malate; complex I is excluded from oxidation and subject to events of reverse electron transfer.

**Figure 10 ijms-24-04045-f010:**
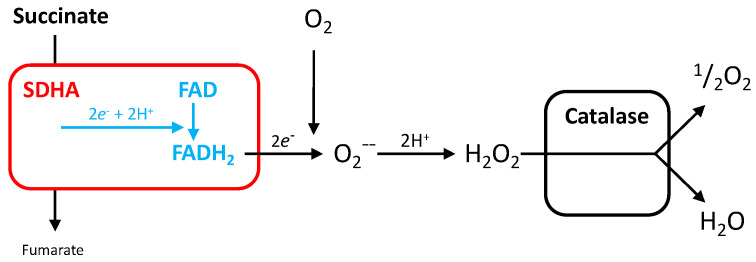
Hypothesis about autonomous SDHA activity. Breaking the link between SDHA and the rest of the SDH subunits would be the result of a loss of SDHB expression. Leakage of electrons from the SDHA FAD/FMN coenzyme would reduce oxygen with the formation of hydrogen peroxide. Catalase activity results in water as the final product for the transfer of two electrons, but with no intervention of MRC. While catalase would prevent H_2_O_2_ accumulation, a sustained elevation in H_2_O_2_ levels is expected.

**Figure 11 ijms-24-04045-f011:**
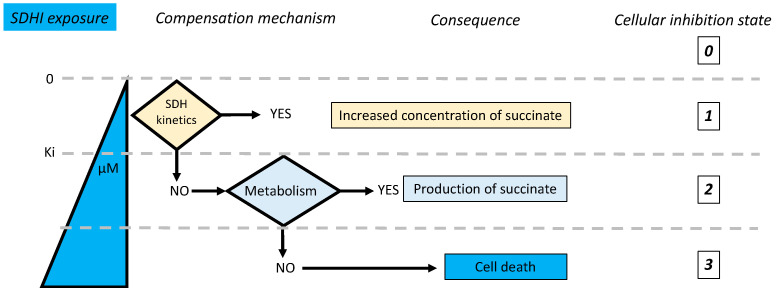
Adaptive responses to increasing levels of inhibition of the SDH. The limit between state 1 (kinetic response) and state 2 (metabolic response) is based on the assumption that inhibition of half or more of the SDH activity would require metabolic compensation with continuous 1/1 replenishment of oxaloacetate in TCA (Appendix A). Too low levels or ablation of SDH activity might be tolerated by some cells but not by others (state 3), leading to cell death.

**Figure 12 ijms-24-04045-f012:**
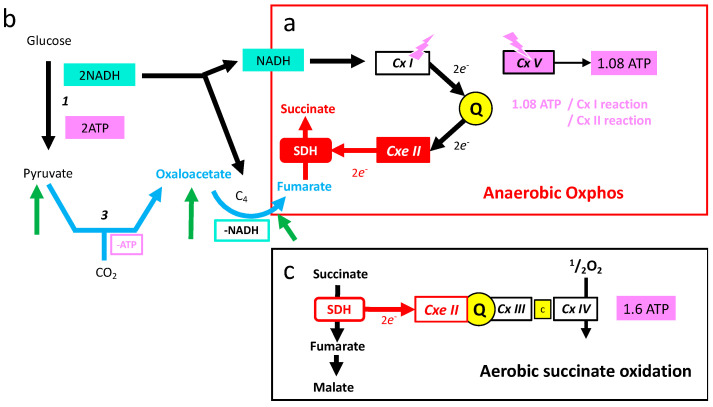
(**a**): Anaerobic MRC with NADH as the electron donor to complex I, and fumarate as the electron acceptor in complex II, (**b**): metabolic generation of oxaloacetate reduced in malate and fumarate; the source could be pyruvate from glucose or from amino acids; it implies CO_2_ assimilation. (**c**): in the presence of oxygen (reperfusion/oxygenated domain), the high succinate concentration causes intense SDH reaction that blocks the entry of electrons from complex I. Therefore, oxidation ends with malate.

**Figure 13 ijms-24-04045-f013:**
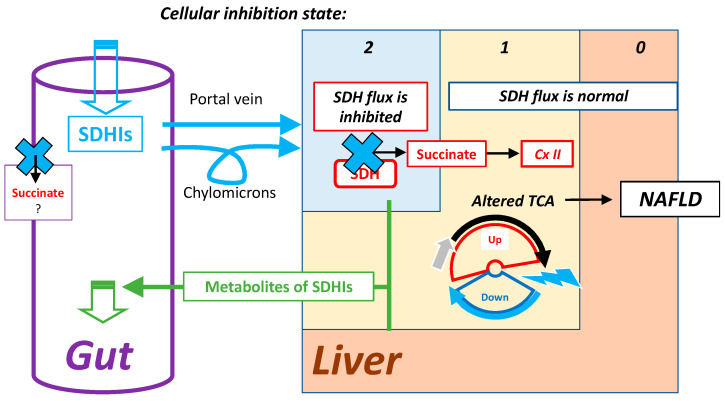
The gastrointestinal tract and liver are the first targets of ingested SDHIs, rest see text.

## Data Availability

Not applicable.

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
