# Peer review of "Inhibition of Succinate Dehydrogenase by Pesticides (SDHIs) and Energy Metabolism"

_ijms, 2023, doi:10.3390/ijms24044045_

Round 1

Reviewer 1 Report

This article is written for an advanced biochemical person.  It is comprehensive and has many things being said.  It may be too long for a less advanced scientist.

Author Response

Thank you for your assessment,

with regard to the audience advanced biochemists/others, it is always a compromise to be made. I hope that readers not familiar with biochemistry would find here an opportunity to understand better few principles expected to ground some of the observations made.

Reviewer 2 Report

Dear Author,

Congratulations for the work. I have read the manuscript entitled "Inhibition of succinate dehydrogenase by pesticides (SDHIs) 2 and energy metabolism."  I think it will be one of the most read review in this subject. It has been written in a clear and descriptive manner. While I was reading I found some very little mistakes that I listed below:

Sincerely Yours,

For the main manuscript:

Line 76 and 485_ ( ) empty parenthesis

For the supplement part:

Page 3, Line 2 and 3_ the word "inhibition" was writeen wrong

Page 5, Figure S5, Line 2_ the word "enlight" can be changed as "enlighten" and aTP must be changed to ATP.

Author Response

Thank you for your assessment of this manuscript.

Correction made

Main manuscript: empty parenthesis were filled with the appropriate text

line 76 : (https://pubchem.ncbi.nlm.nih.gov/)

Supplement part:

P3 letters were missing now corrected for "inhibition"

P5: both modifications were introduced ATP (not aTP) and enlighten (not enlight)

line 485: (see 3.2) 

Reviewer 3 Report

1. In figure 1, there was no information about isoflucypram.

2. As a review, the number of references seems to be insufficient. Some new report about the toxicology of SDHs are missing.

3. The author emphasized that SDHs are toxic to the human body, but the author didn't list the specific dose and phenotype.

4. As far as I know, SDHs are more toxic to aquatic organisms and less toxic to mammals, please discuss why.

Author Response

Thank you for your assessment of this manuscript.

With regard to the modifications requested/comments.

1: the SDHI isoflucypram was introduced as well as another one cyflumetofen. Their introduction does not impact on the rest of the text.

2: references: Two recent references about toxicilogical studies in zebrafish or xenopus have been introduced (now refs 63 and 64) see below. A third reference (now ref 1) on mutations in SDH and resistance to SDHI was introduced too. It was not possible within the limited time to prepare this review to attempt to exhaustivity with regard to references. Moreover, with regard to toxicology of SDHIs, about which it was clearly stated that this was not the topic of the present review (former lines 72-73). This is further explained (now lines 72-77). 

3: It is not completely clear where this statement (toxicity of SDHIs to human body) was present in the manuscript. Or does this refer to the diet issue (§9)? Actually, in the present document human is to be considered within the general case of mammals. What is known/documented in human are the consequences of mutations in SDH genes (chapter 4). Observations with regard to SDHI toxicity in humans should be limited to consequences of accidental exposure to SDHIs used as pesticides. Compilation of these data, if available, would be well beyond my field of expertise.

4. Right, this issue was mentioned at the end of part 7.3 but restricted inappropriately to fishes, revealing a bias towards vertebrates. This part was modified to introduce aquatic organisms opposed to terrestrial organisms. This sensitivity is likely to enhance the response of zebrafish or xenopus to SDHIs/mitochondrial inhibitors and references about toxicological studies were introcuded in thsi paragraph.